# SelectFormer: Private and Practical Data Selection for Transformers

## Abstract

Critical to a free data market is *private data selection*, i.e. the model owner selects and then appraises training data from the data owner before both parties commit to a transaction. To keep the data and model private, this process shall evaluate the target model to be trained over Multi-Party Computation (MPC). While prior work suggests that evaluating Transformer-based models over MPC is prohibitively expensive, this paper makes it practical for the purpose of data selection. Our contributions are three: (1) a new pipeline for private data selection over MPC; (2) emulating high-dimensional nonlinear operators with low-dimension MLPs, which are trained on a small sample of the data of interest; (3) scheduling MPC in a parallel, multiphase fashion. We evaluate our method on diverse Transformer models and NLP/CV benchmarks. Compared to directly evaluating the target model over MPC, our method reduces the delay from thousands of hours to tens of hours, while only seeing around 0.20% accuracy degradation from training with the selected data.

## 1 Introduction

**Data selection & appraisal** In today's ML ecosystem, data and models are often owned by separate parties. Examples include mobile users versus app providers, and small businesses versus marketing firms. As model owners are interested in acquiring data to train their models from data owners, data becomes a commodity to be valued and traded.

In a free data market, no purchase commitment should be required unless both parties reach an agreement, for which pre-purchase assessment is vital. It contains two steps: select the most valuable data; (optionally) appraise the selected data. To model owners, pre-purchase selection is key to cost-effectiveness. Extensive work already showed that datasets are often redundant and noisy Settles (2012); Katharopoulos & Fleuret (2018); Mirzasoleiman et al. (2020); purchase budgets should be spent on data points that are most likely to maximize model accuracy. To data owners, selection allows them to reveal less data in exchange for the same monetary reward, which retains their data ownership and reduces privacy exposure. The need for judicious data selection is further highlighted by that data candidates are often *unlabeled* Hoi et al. (2006); Smith et al. (2018) and have *skewed label distributions* Kaur et al. (2019). This paper targets this problem setting.

**Challenge: private selection** Data selection and appraisal should be private: they should not leak any data and model parameters; if a purchase agreement is eventually reached, the model owner should learn no more beyond the data she is paying for. To select and appraise data, a common algorithm is to run a model's forward passes over data candidates and select based on the outcome (see Section 2 for details). It can be made private by building atop Multi-Party Computation (MPC) Goldreich et al. (1987); Shamir (1979); Yao (1986): both parties (owners) jointly evaluate over the candidate data, learning only the indices of selected data and the quality measurement.

However, evaluating modern models, especially Transformer models, on MPC is expensive. A forward pass of BERT on a batch of 4 requires 3252 communication rounds and over 245 GB data exchanged, taking around 0.76 hours across commodity servers with GPUs. Most of the cost comes from executing nonlinear arithmetics such as softmax, which is pervasive in Transformers. To support Transformer inference over MPC, recent work approximates nonlinearity with cheaper linear operations Li et al. (2022); Dong et al. (2023). For data selection, they are not only too slow (hundreds of GPU hours to select from tens of K data points) but also result in poor data selection as this paper will show.

**Goal & techniques** Our goal is to accelerate private data selection for transformer-based models, e.g. BERT and ViT, while retaining the training performance. Our key insight is that the nonlinearity in Transformers can be fused and evaluated at low dimensions; fortunately, the resultant model evaluation outcomes will be acceptable for the purpose of data selection, which only depends on how

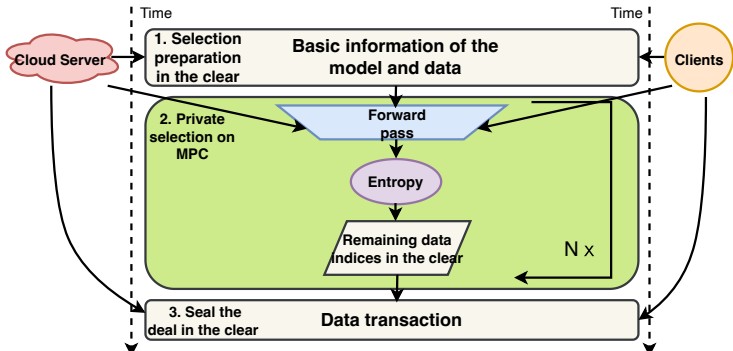

Figure 1: Three stages of our data selection workflow in chronological order.

the outcomes for individual data candidates compare to each other. We present a holistic selection pipeline with three key techniques.

*Nonlinearity emulated by multi-layer perceptron (MLPs).* While existing MPC frameworks treat individual nonlinear operations in isolation (e.g. Newton-Raphson iterations for reciprocal), we fuse adjacent non-linear operators and emulate them altogether with a small MLP. The benefits are twofold. (1) MLPs significantly reduce MPC costs because they not only serve as approximators (converting nonlinear to linear operations) but also dimension reducers, e.g. substituting a 512-dimensional softmax with 2-dimensional MLP. This sets us apart from prior work that approximates a single nonlinear operation (e.g. reciprocal) with an MLP Chen et al. (2022), which does not reduce dimensions and therefore executes much slower. (2) MLPs are data-driven. They are trained atop the distributions of the actual model activations, tailored to the datasets of interest. They only require a small amount of training data, as few as several hundred on average in our experiment.

*Multi-phase selection.* For efficiency, the data/model owners jointly select data in multiple phases. While early phases evaluate cheap selection models to filter most of the data, later phases run more expensive selection models to select from the data that survived from earlier phases. This reduces the total selection delay without compromising accuracy.

*Parallel MPC executions.* To further reduce the selection delay, the data/model owners batch network latency-bound MPC operations and execute them in parallel to network bandwidth-bound MPC operations; they further overlap the MPC communication and computation across different batches. No prior MPC systems exploit such parallelism, to the best of our knowledge.

**Workflow** There are three stages in our proposed workflow: two in the clear and one over MPC. As shown in Figure 1, in the beginning, the model owner and the data owner exchange some basic information in the clear. For instance, the total amount of data and the amount of data that the model owner plans to purchase. Next is the private multi-phase selection which is over MPC. Two parties secretly share encrypted proxy model parameters and data to do a forward pass and get the entropy value. We then rank them and select the top candidates based on their rank. The remaining data points' indices will be reused in the next phase. The comparison outcomes will be revealed and the data indices are in the clear. Data will not be transferred among phases. The final transaction is in the clear. After private data selection, the model owner makes payment for the data it wants and the data owner sends out the data to seal the deal. See Section 4 for details.

**Results** We test SelectFormer with four target models, seven NLP/vision benchmarks, and a variety of purchase budgets (20%–40%). We show that with commodity GPUs and typical Internet, our pipeline can select from 10K–100K data points within tens of hours, one order of magnitude faster than prior work. Using unmodified MPC protocol and framework, we provide a strong privacy guarantee. Our selected data only results in just 0.20% lower accuracy compared to directly evaluating the target model over MPC for selection, a gold method which is however 204×slower.

**Contributions** This work proposes a novel application of MPC – private data selection, and makes it practical on large Transformer models. It achieves so through three new techniques:

• A data-driven MLP approximation technique that is uniquely tailored for data selection, instead of generic inference. Such a novel use of MLP approximation is discovered for the first time.

• Multi-phase selection that progressively increases the selection models' capability and reduces the amount of data candidates.

• Parallel MPC executions that hide the delays of computation and communication rounds behind that of communication data exchange.

## 2 BACKGROUND

### 2.1 ASSUMPTIONS ON SYSTEMS

**Two parties** The model owner possesses a pretrained Transformer model $M_{target}$ and wants to finetune it. The data owner owns the dataset $D$ for selection. The model owner is interested in buying $B$ datapoints ($B < |D|$) from dataset $D$ which will be used to finetune the model $M_t$. The model owner has a private, small validation set, on which she wants to maximize the test accuracy. The two parties agree on $B$ and that they will use MPC for selection.

**Datapoints** We follow a common premise in prior work on data selection: Coleman et al. (2019); Mullapudi et al. (2021); Mahmood et al. (2022), the dataset $D$ is unlabeled and might have an imbalanced class distribution, which warrants careful selection before training. This premise is motivated by real-world situations where data labels are scarce and data owners often lack the motivation, expertise, and authority to label their data Hoi et al. (2006); Smith et al. (2018).

**Threat model** Without alternation, we follow a semi-honest model widely used in prior security research Knott et al. (2021); Dong et al. (2023); Mohassel & Zhang (2017): both parties faithfully follow the MPC protocol but nevertheless attempt to learn each other's private information (model weights and datapoints) through their interactions. Notably, we do not deviate from protocols implemented by well-known frameworks such as Crypten. As such, our privacy guarantee is as strong as the underlying framework.

### 2.2 ALGORITHM FOUNDATIONS

**(1) Maximum entropy selection** Active learning (AL) Settles (2012) selects from unlabeled data to maximize training performance. To appraise an example $x$ for training model $M$, AL runs a forward pass of $M$ over $x$ (or query $x$ with $M$) and computes the prediction entropy: higher entropy implies lower model confidence, hence a higher learning value of $x$. **(2) Proxy models** Querying all examples in $D$ with $M_t$ can be slow. To this end, prior work uses a lightweight model $M_p$ called "proxy" for selection before training $M_t$ with the selected data Coleman et al. (2019). To select valuable data, $M_p$'s prediction should resemble that of $M_t$; hence $M_p$ is often created as a miniature version of $M_t$: examples include ResNet-18 as proxy for ResNet-50 Coleman et al. (2019). We use simplified Transformers as proxy models with fewer layers and attention heads.

Figure 2: Transformers over MPC incurs high communication and computation overhead. Showing one forward pass of one layer (12 heads) over a batch of 5 (maximum allowable on our GPU). Hardware: Quadro RTX 4000. MPC framework: Crypten Knott et al. (2021).

### 2.3 TRANSFORMERS, MPC, AND THEIR OVERHEAD ANALYSIS

**Transformers** We focus on transformer-based models Vaswani et al. (2017). Each layer of these models features a Multihead Attention paired with FeedForward. Both are succeeded by layer normalization and a residual connection. Within the Multihead Attention, operations proceed sequentially from QKV linear operation to the attention mechanism, and finally to an attention output.

**MPC setting: 2PC** We assume the most common MPC setting: secret sharing under two-party computation (2PC) Yang et al. (2019). The model owner randomly decomposes $x$ into two shares, $x_1$ and $x_2$ (such that $x_1 + x_2 = x$), retains $x_1$, and convey $x_2$ to the data owner. Analogously, the data owner fragments $y$ into $y_1$ and $y_2$ and transfers $y_2$ to the model owner. These shares individually disclose no information about the original numbers, $x$ and $y$. Reconstruction of a number is possible when both parties exchange their shares and sum them up. To compute $z = x + y$, each party add

their shares of $x$ and $y$, yielding $z_1 = x_1 + y_1$ and $z_2 = x_2 + y_2$. Both parties can retrieve $z$ by adding $z_1$ and $z_2$. For multiplication, The parties can offline generate a random triple called Beaver triple Beaver (1992), $a$, $b$, and $c$ (such that $ab = c$). The elements of this triple are then partitioned and disseminated amongst the parties, akin to $x$ and $y$. The model owner calculates $\epsilon_1 = x_1 - a_1$, $\delta_1 = y_1 - b_1$ while the data owner computes $\epsilon_2 = x_2 - a_2$, $\delta_2 = y_2 - b_2$. The parties jointly reconstruct $\epsilon$ and $\delta$. Subsequently, the model owner computes $p_1 = c_1 + \epsilon y_1 + \delta x_1 + \epsilon \delta$ and the data owner calculates $p_2 = c_2 + \epsilon y_2 + \delta x_2$, enabling the retrieval of $p = xy$. Throughout this procedure, all intermediate computations, including $\epsilon_1$, $\epsilon_2$, $\delta_1$ and $\delta_2$ maintain the confidentiality of $x$, $y$ and $p$, precluding any information leakage.

**Major overheads** comes from that MPC lacks native support for nonlinear operations, such as softmax, logarithmic, and exponential, which are pervasive in Transformers. In response, MPC frameworks provide corresponding linear approximations. Unfortunately, the overhead is still high. Figure 2 shows the cost of the operations for *one* transformer block. As shown in the table, the cost of nonlinear arithmetic dominates. Notably, softmax contributes 81.9% communication data and 142 communication rounds. The overheads become prohibitive for data selection, which needs to evaluate Transformers on tens of thousands of data points over MPC. We show this in Section 5.

## 3 RELATED WORK

**Data selection and appraisal** Our work builds on the idea of using a miniature version (proxy) of the target model as the selector Coleman et al. (2019) Yet, proxies alone are still too expensive for MPC as we will show in Section 5. Xu et al. (2022) investigates data appraisal over MPC. Contrasting to us, they assume labeled data (a strong assumption in practice), which allows them to appraise data via forward influence functions. Unlike us focusing on Transformer models, they only demonstrated logistic regression tasks. Mindermann et al. (2022) introduces RHO loss that quantifies by how much data candidates would reduce the loss on a small holdout set if the model were to train the candidates. They then select data based on RHO loss. Their methods inspire our bootstrap dataset. However, they rely on that data candidates must be labeled, whereas we focus on a more challenging setting – selecting unlabeled data.

**MPC inference of neural networks** Existing research has developed discretized DNN training with a customized two-party protocol Agrawal et al. (2019) and MPC-friendly approximations to speed up CNN computation in MPC. Chou et al. (2018) develops an optimization framework that minimizes the approximation error of order 2 polynomial to ReLU. Mishra et al. (2020) alleviates this problem by using a set of carefully designed heuristics along with Neural Architecture Search (NAS). Mohassel & Zhang (2017) propose an approximation to softmax by replacing exponential with ReLU functions on high-dimension inputs which are still costly.

MPCFORMER Li et al. (2022) and PUMA Dong et al. (2023) directly address efficient Transformer inference over MPC. MPCFORMER's key idea is that approximate nonlinear operations in the student model and then distill it; PUMA proposes new approximations for faster secure inference in a three-party setting. Compared to us, they miss the key opportunities of reducing nonlinearity dimensions and using learned approximators (MLPs). Furthermore, Hence, MPCFORMER's model distillation approach is ill-suited to data selection, where labeled data for distillation is lacking. As Section 5 will show, they perform poorly in data selection.

## 4 METHOD

### 4.1 HIGH-LEVEL WORKFLOW

**Pre-selection bootstrap:** Prior to the selection, the two parties exchange meta-information *in the clear*: the purchase budget $B$, the secure computation framework, the target model architecture type, pre-processing methods, and the metadata about the $S$, which includes the number of data points.

The model owner purchases bootstrap data $S_{boot}$ which constitutes a small fraction of the purchase budget $B$. The data owner randomly samples $S_{boot}$ from $S$; the purchase requires no selection and incurs no MPC overhead. The model owner will use $S_{boot}$ to generate proxy models $\hat{M}_{1..N}$, as will be described below.

**Multipass selection** For efficiency, the selection process is a progressive sieve: earlier phases run smaller selector models for quickly filtering a majority of redundancy, whereas later phases run larger selector models for slower, more precise selection.

Concretely, both parties jointly execute $N$ selection phases over MPC. A phase $i$ downselects dataset $S_{i-1}$ to the set $S_i$ with a selectivity $\alpha_i = S_i/S_{i-1}, \alpha_i < 1$.

In phase $i$, the model owner queries the dataset $S_{i-1}$ with a proxy model $\hat{M}_i$. The query is executed as a forward pass of $\hat{M}_i$ on $S_{i-1}$ on MPC. The forward pass computes the prediction entropy values, which are encrypted. Next, both parties jointly find the *indices* of $|S_i|$ highest entropy values: execute the QuickSelect algorithm with $O(|S_{i-1}|)$ pairwise comparisons between entropy values; the comparison is over

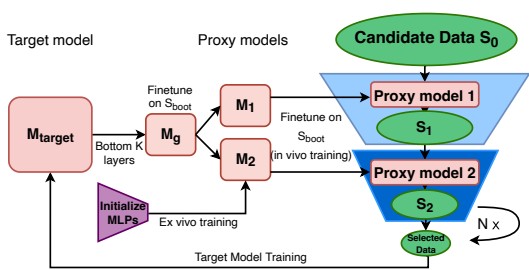

Figure 3: Our high-level workflow. Left: proxy model generation. Right: multi-phase selection.

MPC, each taking 8 communication rounds and transferring 432 bytes. A comparison does not reveal its inputs (entropy values) but only the binary outcome. Finally, the indices of data points with the highest prediction entropy constitute the input $S_i$ for the next phase. $N$ phases result in the final dataset $S_N$. The model owner may offer to buy $S_N$, or request an appraisal of $S_N$ before buying. For appraisal, both parties jointly compute an average entropy over $S_N$ and reveal the average entropy. If the average entropy is sensitive, both parties can jointly compute if the (encrypted) average exceeds a pre-defined threshold and only reveals the one-bit outcome.

**Privacy guarantees** The following information is kept private: dataset $S$; parameters of $M_{target}$ and $\hat{M}_{1..N}$; the prediction outcome and entropy. The following is revealed: the *rank* of prediction entropy; the architecture (operations) of $\hat{M}_{1..N}$; the number of data points and budget.

### 4.2 GENERATING PROXY MODELS IN DETAIL

A proxy model $\hat{M}_i$ is characterized by $< l_i, w_i, d_i >$: $L_i$ transformer layers with $W_i$ attention heads each (width), in which the nonlinear modules are substituted by MLPs with hidden dimension $d_i$ and the GeLU functions are substituted by ReLU functions. FFN is also removed from proxy models. As shown in Figure 3, the model owner generates proxy models $\hat{M}_{1..N}$ as follows:

• Extract a sub-model $M_g$ from $M_{target}$. $M_g$ comprises $L$ bottom layers from $M_{target}$ where $L = max(l_{1..N})$, with all the weights copied over. $M_g$ serves as the backbone for $\hat{M}_{1..N}$.
• Finetune $M_g$ on the bootstrap data $S_{boot}$. This serves two purposes: (1) collecting sample input/output of $M_g$'s nonlinear modules, which will be used to train the MLPs that substitute these nonlinear modules; (2) roughly adapting $M_g$ to a sample from the dataset $S$. Initialize $\hat{M}_{1..N}$ by pruning the width and depth of $M_g$.
• *Ex vivo* MLP training. For each transformer block in a proxy model, randomly initialize three MLPs. Train them separately on the input/output of nonlinear modules collected from the previous step. More training data following the same distribution as the input/output will be randomly generated as data augmentations. See subsection 4.3 for details.
• *In vivo* MLP training. Substitute the nonlinear modules in $\hat{M}_i$ with the MLPs trained in the previous step. Further finetune the resultant model $\hat{M}_i$ on $S_{boot}$ end-to-end. This co-tunes MLPs (the approximate portion of $\hat{M}_i$) and the remaining exact portion.

The model owner schedules the selection by setting $\{< l_i, w_i, d_i >\}_{i=1..N}$ for $N$ phases. As described above, the principle of schedule is progressive: increasingly higher $l/w/d$ for later phases. Given a budget $B$, SelectFormer determines the schedule via offline grid search. See Section 5

### 4.3 APPROXIMATION MLPS FOR NON-LINEAR OPERATIONS

Our theoretical foundation is Hornik et al. (1989): MLPs are able to approximate any function at any desired degree of accuracy, as long as the function is continuous on a closed and bounded subset of $R^n$. A transformer's nonlinear modules satisfy this condition Goodfellow et al. (2016). We use

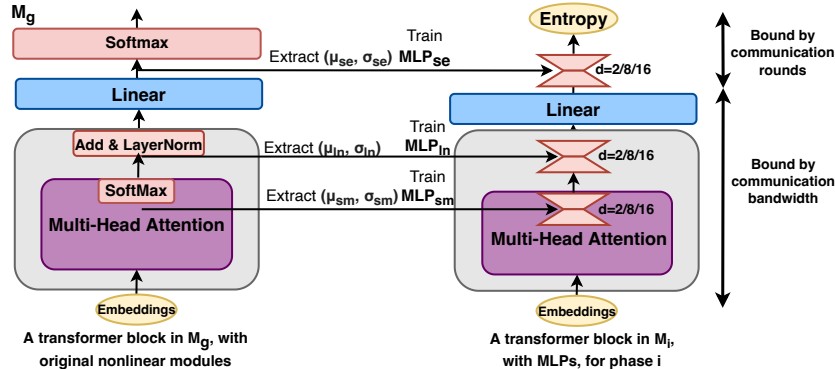

Figure 4: Training MLPs for substituting the non-linearity in Transformer models

standard MLPs, each comprising one ReLU layer sandwiched between two linear layers. Of a proxy model, MLPs substitute the following nonlinear modules as shown in Figure 4:

• *Softmax in the attention module*. At each transformer layer, a substitute MLP maps along the last dimension of the input and has the same input/output shape as the original softmax operation.

• *LayerNorm in the attention module*. At each transformer layer, a substitute MLP emulates the reciprocal operation in LayerNorm. Of the LayerNorm, its learnable weights and bias in the affine functions are loaded from the original LayerNorm layers in $M_g$. Its numerator is computed directly on MPC because the total number is a constant and it needs just cheap summation and multiplication.

• *Softmax over logits and the subsequent entropy computation*. At the top of the model, a substitute MLP emulates the two modules combined. The MLP input shape is the same as that of softmax. The output will be the entropy itself.

A proxy model of layer $l$ comprises $2l + 1$ MLPs in total; they have the identical hidden dimension albeit different weights. Different proxy models may have MLPs of different hidden dimensions, set as part of the schedule as described above.

**Training MLPs**  MLPs need to be trained on the sample input/output of the nonlinear modules being substituted. While we can use such input/output observed in finetuning $M_g$ over bootstrap data ($S_{boot}$), the data amount is inadequate: randomly sampled $S_{boot}$ must be small (3521 datapoints on average in our experiments) in order to save the budget for private selection.

We address the problem with the empirical observation that inputs to a nonlinear module largely follow a parametric Gaussian distribution Chen et al. (2022). As such, from the actually observed inputs, the model owner estimates the Gaussian distribution parameters $\langle \mu, \sigma \rangle$ and uses such parameters to synthesize training inputs and outputs. For a given nonlinear module in $M_g$, an instance of $\langle \mu, \sigma \rangle$ is estimated and one dataset (5.12 million data points) is synthesized once, which is used to train the MLPs that substitute this particular nonlinear module in all proxy models $M_1..N$. There are three synthesized datasets: $S_{sm}$, $S_{ln}$, and $S_{se}$. This is shown in Figure 4. Note that $M_1..N$ with MLPs inserted will be further finetuned as described above. The MLP training is very cheap compared with the selection over MPC, which can be measured in minutes.

### 4.4 IO SCHEDULING

Our proxy models perform a sequence of matrix multiplication and ReLU. When operating on high-dimensional inputs, they are bound by the network bandwidth; after being projected into lower dimensions, the operations succeeding the activation functions are bound by network latency. To this end, (1) SelectFormer stacks and coalesces the latency-bound modules from multiple batches, reducing the number of communication rounds and improving the computation throughput. (2) Furthermore, SelectFormer exploits computation and communication parallelism across batches, executing a module whenever its needed resources become available. For instance, while data exchange is occurring for one batch, computations for the subsequent batches can be performed concurrently. Such co-execution is only limited by data dependencies and the available memory of a party to hold operation inputs.

| | DistilBERT | | | | | BERT | | | | | ViT-small | | ViT-base | |
|---|---|---|---|---|---|---|---|---|---|---|---|---|---|---|
| | SST2 | QNLI | QQP | AGNEWS | YELP | SST2 | QNLI | QQP | AGNEWS | YELP | Cifar10 | Cifar100 | Cifar10 | Cifar100 |
| Ours | 82.34±1.01 | 71.46±1.06 | 71.53±0.77 | 88.49±0.55 | 60.18±0.20 | 84.59±0.95 | 73.68±1.64 | 72.73±0.60 | 89.04±0.43 | 61.33±0.40 | 96.46±0.25 | 69.11±1.16 | 96.61±0.17 | 68.42±1.44 |
| Random | 79.08±1.65 | 70.34±1.74 | 66.18±3.54 | 86.43±0.66 | 58.09±0.60 | 79.58±1.83 | 73.56±1.35 | 68.84±0.74 | 87.85±0.54 | 59.16±0.43 | 94.69±0.37 | 43.99±0.93 | 95.63±0.50 | 63.21±1.78 |
| (vs. Ours) | -3.26 | -1.12 | -5.35 | -2.06 | -2.09 | -5.01 | -0.12 | -3.89 | -1.19 | -2.17 | -1.71 | -25.12 | -0.98 | -5.21 |
| Oracle | 82.32±0.68 | 71.44±1.35 | 71.51±1.51 | 89.19±0.27 | 59.93±0.16 | 86.06±0.79 | 74.62±1.39 | 73.08±1.86 | 89.74±0.38 | 60.80±1.33 | 96.87±0.17 | 68.40±1.71 | 96.94±0.23 | 67.85±1.35 |
| (vs. Ours) | -0.02 | -0.02 | -0.02 | +0.70 | -0.25 | +1.47 | +0.94 | +0.35 | +0.70 | -0.53 | +0.41 | -0.71 | +0.33 | -0.57 |

Table 1: *Ours* are consistently higher than *Random* across all models and benchmarks and are close to *Oracle* (gold accuracy). Accuracies and standard deviations are gotten by 5 runs.

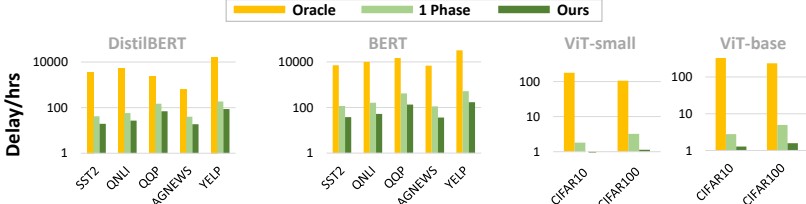

Figure 5: Our method reduces the end-to-end selection delays by two orders of magnitude. As in section 4, 1 phase selection selects data using a proxy model with a hidden dimension of 16.

## 5 EVALUATION

### 5.1 EXPERIMENT SETUP

**Models & datasets** For NLP, we choose target models as BERT (12 layers) and DistilBERT (6 layers), with 12 attention heads and a hidden dimension of 768. For vision, we chose the target model as ViT, with the same architecture as BERT. We chose benchmarks with at least tens of thousands of data points which warrant selection. Following prior work Xu et al. (2022) to construct the training set with imbalanced labels, we remove some data points from the original benchmarks. Our imbalanced NLP benchmarks include binary classification from GLUE Wang et al. (2018): SST2 (42K), QNLI (58K), and QQP (149K); and multilabel classification Zhang et al. (2015): AG news (40K) and Yelp review full (188K). Our vision benchmarks are two multilabel benchmarks CAFAR-10 (10K) and CIFAR-100 (6K). Note that we do not change the test set.

We implement our scheme based on Crypten Knott et al. (2021), a popular MPC framework. We run data/model owners on separate GPUs (Nvidia Quadro RTX 4000) with Intel(R) Xeon(R) Silver 4210 CPU; we control the network bandwidth (100 MB/sec) and latency (100 ms) between them to emulate a wide-area network condition. We report: (1) the test accuracy of the target model, after being trained with the selected data; (2) the delay of the data selection process over MPC.

**Baselines** We evaluate our method against the following baselines.

1. *Random* selects data randomly from the data owner. It incurs zero MPC overhead.

2. *Oracle* queries data using the target model. Under our framework (Section 2), we consider that it leads to gold test accuracy. Yet, it incurs prohibitive MPC overhead as will be shown.

3. *MPCFORMER* Li et al. (2022) is a closely related project that minimizes the cost of evaluating transformers over MPC. We implement this baseline to adopt MPCFORMER's key optimizations for evaluating proxy models: linear operators for approximating nonlinearity and model distillation for recovering accuracy. For a fair comparison, *MPCFORMER* uses the same proxy model architecture as ours, and initializes the proxy models in the same way; runs distillation on the same amount of bootstrap data. Note that we cannot directly compare to PUMA Dong et al. (2023), which is designed under 3 compute parties. and three computing parties.

### 5.2 END-TO-END RESULTS

**Selection delays** Our method significantly reduces the delays. Compared to *Oracle*, our method's delay is lower by two orders of magnitude as shown in Figure 5. For instance, to select 20% from 42K data points (SST2 benchmark, DistilBERT model), our experiment takes around 20 hours whereas *Oracle* would take 3740 hours.

**Selection efficacy** Our method effectively selects training data that best boosts the test accuracy. Table 1 zooms in on a typical purchase budget (20%). Compared to *Random*, our test accuracy is consistently higher across all NLP and CV benchmarks. The benefit is particularly pronounced on challenging tasks such as CIFAR-100, on which our accuracy is higher by 25.11%/5.21% on ViT-small/base. Compared to *Oracle*, our test accuracy is comparable: only 0.08% – 0.58% lower for DistillBERT and BERT, and even 0.13% and 0.15% higher for ViT-base and ViT-small.

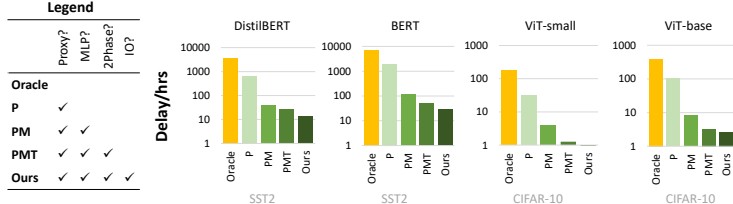

Figure 6: Across a variety of budgets, *Ours* consistently outperforms *Random* and are comparable with *Oracle* (gold accuracy). Target model: DistilBERT.

Figure 7: Delay reduction by our techniques. The results also show that using proxy models alone (variant "P" in the figure, Coleman et al. (2019)) is not enough.

Across different purchase budgets of 25%, 30%, and 40%, our method also leads to strong accuracy. Compared to *Random*, our accuracy is 2.23% higher on average, and up to 5.14%. Compared to *Oracle*, our average accuracy is only 0.91% lower on average, and as low as 0.01%. See Table 6 in the Appendix for detailed results.

|  | BERT Accuracies/% | | | Delay/hrs | | |
|---|---|---|---|---|---|---|
|  | SST2 | QNLI | QQP | SST2 | QNLI | QQP |
| MPCFORMER | 50.92 | 50.54 | 36.82 | 279 | 388 | 998 |
| Ours | 84.59 (+33.67) | 73.68 (+23.14) | 72.73 (+35.91) | 37 (-242) | 51 (-337) | 131 (-867) |

Table 2: Compared to MPCFORMER Li et al. (2022), our method results in much higher test accuracy and much lower delays. Focus on GLUE benchmarks which were also evaluated and reported by MPCFORMER. Target model: BERT. Proxy model: 1 transformer layer with 1 heads + 3 transformer layers with 12 heads.

From a different angle to view our efficacy: to reach a given accuracy, our method demands a much lower budget. As shown in Figure 6, to be on par with our test accuracy on a budget of 20%, *Random* would need a budget of 70% (BERT on QNLI), a budget of 90% (DistilBERT on SST2), and a budget of 100% (DistilBERT on YELP and BERT on YELP/SST2).

## 5.3 COMPARISON WITH MPCFORMER

As shown in Table 2, our method simultaneously delivers much higher test accuracy (by 23%–36%) and much shorter (by 7×) selection delays. The reasons are twofold. (1) *MPCFORMER*'s longer delays come from that it approximates individual nonlinear operators with linear counterparts; it does not reduce model dimension as we do. (2) *MPCFORMER*'s lower accuracy comes from that its proxy models come from the distillation of the target model, a process requiring ample, representative training data. This is at odds with data selection: $S_{boot}$ is the only data available for distillation; it has a small size and skewed labels. The skewness propagates from $S_{boot}$ to proxy models, to selected data, and to the target model. As a result, the trained target model is biased towards predicting the majority class in the training set. This results in poor accuracy if the test and train sets have different majority classes, sometimes even worse than random guess (e.g. QQP in Table 2)

## 5.4 ABLATION STUDY

**Crypten incurs minor accuracy loss** We validate that our choice of the Crypten framework Knott et al. (2021), which executes ML on a finite ring, introduces minor accuracy loss. On SST2 with the default budget (5% for $S_{boot}$; 20% total), private selection fully executed on Crypten results in test accuracy of 83.37%, which is even slightly (0.5%) higher than executing on PyTorch.

**MLP for non-linearity** (1) Delay. Our use of MLP reduces the selection delay by two orders of magnitude, e.g. from 3740 hours to around 20 hours for DistilBERT on SST2. This is shown in Figure 7, by the difference between *P* and *PM*. (2) Accuracy. The use of MLP incurs minor degradation in the test accuracy. The results are shown in Table 3, by the difference between *Ours* to *NoApprox*. On SST2, ours is 0.78% higher with DistilBERT and only 0.46% lower with BERT. On AGNEWS, ours are just 1.24% lower with DistilBERT and 0.93% lower with BERT. On QQP, it's 1.21% and 0.05% higher respectively. Among the three MLPs, softmax in the attention module

| | Emulated w/ MLP? | | | DistilBERT Accuracy | | | BERT Accuracy | | |
|---|---|---|---|---|---|---|---|---|---|
| | $MLP_{sm}$ | $MLP_{se}$ | $MLP_{ln}$ | SST2 | QQP | AGNEWS | SST2 | QQP | AGNEWS |
| Ours | ✓ | ✓ | ✓ | 82.80±0.98 | 71.63±0.91 | 87.75±1.00 | 84.33±1.30 | 72.31±1.21 | 88.37±1.58 |
| NoAttnSM | | ✓ | ✓ | 82.48±1.27 (-0.32) | 71.85±1.11 (+0.22) | 89.58±0.43 (+1.83) | 85.14±0.46 (+0.81) | 74.15±0.56 (+1.84) | 89.55±0.66 (+1.18) |
| NoAttnLN | ✓ | ✓ | | 81.83±0.78 (-0.97) | 71.68±0.44 (+0.05) | 87.20±1.49 (-0.55) | 83.95±0.73 (-0.38) | 72.76±0.94 (+0.45) | 89.08±0.47 (+0.71) |
| NoApprox | | | | 82.02±0.80 (-0.78) | 70.42±1.00 (-1.21) | 88.99±0.50 (+1.24) | 84.79±1.00 (+0.46) | 72.26±0.91 (-0.06) | 89.30±0.45 (+0.93) |

Table 3: Using MLPs to emulate Transformer nonlinearity results in minor accuracy degradation

| Totl Phs | MLP dims | # proxy layers | | Selectivity | DistilBERT Accuracy | | BERT Accuracy | |
|---|---|---|---|---|---|---|---|---|
| | | DistilBT | BERT | | SST2 | QQP | SST2 | QQP |
| 1 | 16 | 1 | 1 | 100%*→20% | 82.91 | 71.28 | 83.49 | 64.48 |
| 2 | 2→16 | 1→1 | 1→3 | 100%*→30%→20% | 82.00 (-0.91) | 71.62 (+0.34) | 85.21 (+1.72) | 73.07 (+8.59) |
| 3 | 2→8→16 | 1→1→1 | 1→3→3 | 100%*→50%→30%→20% | 83.37 (+0.46) | 72.90 (+1.62) | 84.52 (+1.03) | 73.48 (+9.00) |

*: include bootstrap data

Table 4: Our multi-phase selection improves accuracy in general. Three example schedules (with 1, 2, and 3 total phases, respectively) and their test accuracies are shown.

shows a higher impact on accuracy than the other two. As shown in Table 3 and across all the datasets, by the difference between *Ours* to *NoAttnSM*. Using softmax approximation MLP is just 0.75% lower with DistilBERT and 0.99% lower with BERT on average; by the difference between *Ours* to *NoAttnLN*, our approximations of LayerNorm only decreases the performance by 0.76% with DistillBERT and 0.16% higher with BERT on average. This is justified by the higher cost that our AttentionSM reduces data communication by 42×, while AttentionLN only reduces 8.25×.

**Efficacy of multi-phase selection (MPS)** We compare MPS to single-phase selection (SPS), in which the proxy model is the same as the one used in the last phase of MPS; it has all our other optimizations. (1) MPS reduces delays significantly. Compared to SPS, the two-phase selection reduces the total delay by 33%–61% across all the benchmarks. Because MPS runs a much smaller proxy model in phase 1 to filter most datapoints. The advantage is more pronounced for large models such as BERT, where the efficiency gap between the proxy models in phase 1 is even larger. (2) MPS improves accuracy moderately. We focus on two representative benchmarks SST2 and QQP (binary and multi-label classification). The results are in Table 4. Going from one phase to two/three phases results in tangible accuracy gain (around 1%); notably BERT sees accuracy gain as high as 1.72% (SST2) and 8.59% (QQP). We attribute the gain to MPS being more selective for the finally selected datapoints, which have been sieved through multiple proxy models, small and big.

**Sizes of proxy models** Multi-layer proxy models increase the selection delay while may improve the test accuracy. Our experiments show that: for small target models on challenging benchmarks (e.g. ViT-small on CIFAR-100), the test accuracy from a three-layer proxy outperforms a one-layer proxy significantly (by 17.33%); on other benchmarks, 1+ layers only result in a minor increase in the test accuracy. On CIFAR-100, we find that the accuracy of a three-layer ViT-small proxy model is 17.33% higher than a one-layer and a three-layer ViT-base proxy model is 11.32% higher than a one-layer with 66.6% lower efficiency. Their accuracies are 0.26% higher than Oracle's on average, but with 4× higher efficiency. It is worth trading some extent of efficiency for much higher accuracy by a multi-layer proxy model, especially on difficult datasets like CIFAR-100.

**MLP hidden dimensions** To decide $d\_i$, we conduct a grid search to find the best values, which are included in our selection schedules shown in Table 4. A higher dimension like 16 has good accuracy on different datasets while not incurring much more latency. Multi-phase selection can reduce lots of latency, hence, we pick hidden dimension combinations that can bring more accuracy gain. For the 2 Phase, we choose (2, 16) which has the highest accuracy on larger datasets and target models, such as BERT on QQP. For the 3 Phase, we choose (2, 8, 16). It has the highest accuracy with DisitilBERT on SST2, with BERT on QQP, and suboptimal accuracy on other experiments.

**IO scheduling** This optimization reduces the end-to-end selection delays by 1.3-1.4×, as shown in Figure 7, the difference between *Ours* and *PMT*. The reduction comes from overlapping the MPC computation and the communication rounds. their communication latency is hidden. The reduction depends on hardware: device computation bandwidth and the network latencies.

## 6  CONCLUDING REMARKS

This paper addresses the private selection of imbalanced, unlabelled data. We propose data selection on MPC, preserving privacy while evaluating data. Overcoming computational challenges, we introduce a data-aware MLP approximation for non-linear modules, tailored to different datasets for reduced latency. Our multi-phase selection mechanism enhances proxy models' efficiency across diverse benchmarks.

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

## 7 APPENDIX

### 7.1 MLP APPROXIMATION IS TIED TO DATA SELECTION

Inspired by the outstanding performance of our MLP approximations for nonlinear modules, one question was raised: is the good performance of MLP approximations because of its inference capability? Our further experiments show that our MLP approximation is tied to data selection, instead of general model inference.

Following Li et al. (2022), we replace three nonlinear modules in a BERT-base model with our approximations: Attention Softmax, Attention LayerNorm, and Feedforward Network Layernorm. There are 3*12 = 36 MLPs in the BERT model. In the experiments, two LayerNorm approximations are always data-aware, while there are both data-aware and fixed approximations of Softmax because of varying attention masks. The BERT with MLP approximation will be fine-tuned on the benchmark before inference.

Three versions of attention softmax approximations achieve no better than random guess performances: (1) Using fixed MLP for attention softmax and changing the mask value to -3 instead of a very negative value for simpler distribution. Our BERT with approximation achieves only 52.92% accuracy. (2) Using data-aware attention softmax approximation, which zeros out masked values on MLP outputs, has 49.08% accuracy. If we normalize the remaining values to [0, 1], the accuracy remains 49.08%. To better understand the influence of each nonlinear approximation, we did an ablation study that keeps just one kind of MLPs (12 of them) in BERT. However, none of them achieve better than 50.92% accuracies. We further notice that adding just one attention softmax MLP to the model has no impact on the inference accuracy. But adding one layernorm MLP will degrade the accuracy by 0.85% on average. These results show that having just one MLP will hurt the inference performance a lot; having 12 or even 36 MLPs will degrade model performance drastically. (3) Removing the attention mask to make MLP data-aware. It always has 50.92% accuracies, no matter with all three approximations or just one approximation.

The poor inference performance and good data selection performance show that the MLP approximation is specifically suitable for data selection while impractical for model inference directly.

## 7.2 ADDITIONAL EXPERIMENTS RESULTS

| # phases | #MLP dims | DistilBERT | | BERT | |
|---|---|---|---|---|---|
| | | SST2 | QQP | SST2 | QQP |
| 1 | 2 | 83.94 | 71.90 | 84.52 | 64.06 |
| | 4 | 85.32 | 70.06 | 83.94 | 63.13 |
| | 8 | 82.11 | 73.43 | 86.58 | 64.20 |
| | 16 | 82.91 | 71.28 | 83.49 | 64.48 |
| | Average | 83.57 | 71.67 | 84.63 | 63.97 |
| 2 | (2, 2) | 82.91 | 72.30 | 84.86 | 72.79 |
| | (2, 8) | 82.45 | 69.84 | 81.88 | 69.30 |
| | (2, 16) | 82.00 | 71.62 | 85.21 | 73.07 |
| | (4, 16) | 82.57 | 72.19 | 83.49 | 72.56 |
| | Average | 82.48 | 71.49 | 83.86 | 71.93 |
| 3 | (2, 2, 2) | 81.77 | 70.92 | 83.37 | 71.67 |
| | (2, 2, 8) | 80.62 | 71.85 | 85.55 | 71.31 |
| | (2, 2, 16) | 82.11 | 72.35 | 84.75 | 73.08 |
| | (2, 8, 8) | 82.57 | 70.79 | 85.21 | 72.77 |
| | (2, 8, 16) | 83.37 | 72.90 | 84.52 | 73.48 |
| | (2, 16, 16) | 81.65 | 71.46 | 86.01 | 71.75 |
| | Average | 82.01 | 71.71 | 84.90 | 73.47 |

Table 5: Multi-phase selection is able to significantly reduce runtime and maintain comparable performance. On QQP, three-phase selection achieves 9.5% higher accuracy than using one phase. main results: 16, (2,16), (2,8,16)

| Benchmarks | SST2 | | | QNLI | | | QQP | | | AGNEWS | | | YELP | | |
|---|---|---|---|---|---|---|---|---|---|---|---|---|---|---|---|
| DistilBERT | Ours | SelectviaFull | Random | Ours | SelectviaFull | Random | Ours | SelectviaFull | Random | Ours | SelectviaFull | Random | Ours | SelectviaFull | Random |
| 20% | 82.80±0.98 | 82.32±0.68 | 79.08±1.65 | 71.12±1.52 | 71.44±1.35 | 70.34±1.74 | 71.63±0.91 | 71.51±1.51 | 66.18±3.54 | 87.75±1.00 | 89.19±0.27 | 86.43±0.66 | 60.04±0.16 | 59.93±0.16 | 58.09±0.60 |
| 25% | 82.75±1.41 | 83.97±1.01 | 77.61±0.82 | 71.04±1.56 | 72.72±1.09 | 69.95±0.93 | 72.04±0.56 | 73.18±1.29 | 68.67±1.11 | 88.06±1.56 | 89.35±0.38 | 87.67±0.46 | 60.38±0.13 | 60.63±0.20 | 58.43±0.20 |
| 30% | 83.35±0.59 | 83.97±0.77 | 78.67±0.23 | 71.10±1.05 | 72.87±0.92 | 70.72±0.94 | 72.10±0.73 | 74.04±0.52 | 69.46±0.58 | 88.28±1.71 | 89.50±0.32 | 88.02±0.72 | 60.86±0.21 | 61.14±0.21 | 58.90±0.27 |
| 40% | 83.30±1.06 | 83.88±1.32 | 80.21±0.81 | 72.03±1.32 | 72.92±1.06 | 70.92±0.19 | 73.43±1.23 | 73.97±0.61 | 69.76±0.54 | 89.72±0.28 | 89.73±0.26 | 87.81±0.66 | 60.92±0.27 | 61.15±0.24 | 59.13±0.49 |

Table 6: Our methods are robust against the increase in purchase budget. We have consistently better performance than Random by a large margin and achieve comparable performance with Oracle, up to 40% budget.

| | DistilBERT | | | | | BERT | | | | |
|---|---|---|---|---|---|---|---|---|---|---|
| | SST2 | QNLI | QQP | AGNEWS | YELP | SST2 | QNLI | QQP | AGNEWS | YELP |
| 40% | 80.21%±0.81 | 70.92%±0.19 | 69.76%±0.54 | 87.81%±0.66 | 59.13%±0.49 | 80.99%±1.74 | 73.15%±0.99 | 71.07%±0.85 | 88.36%±0.67 | 59.85%±0.55 |
| 50% | 79.79%±1.22 | 70.34%±0.70 | 71.07%±0.27 | 88.48%±0.44 | 59.59%±0.23 | 80.85%±0.73 | 72.98%±1.58 | 71.59%±1.20 | 88.92%±0.35 | 60.06%±0.30 |
| 60% | 80.41%±0.80 | 71.47%±1.06 | 71.76%±1.08 | 88.84%±0.63 | 59.73%±0.18 | 82.39%±0.95 | 73.01%±0.98 | 72.28%±0.75 | 88.80%±0.86 | 59.73%±0.88 |
| 70% | 80.53%±1.68 | 71.63%±0.58 | 72.92%±0.47 | 89.02%±0.20 | 59.74%±0.32 | 82.78%±0.41 | 74.02%±1.35 | 72.47%±0.65 | 88.73%±0.37 | 60.46%±0.33 |
| 80% | 81.08%±0.94 | 71.40%±0.79 | 72.69%±1.03 | 89.42%±0.44 | 59.98%±0.24 | 83.00%±1.04 | 75.29%±0.76 | 72.90%±0.44 | 89.42%±0.29 | 60.56%±0.51 |
| 90% | 82.50%±1.00 | 72.39%±0.67 | 73.73%±1.08 | 89.45%±0.35 | 59.96%±0.50 | 82.73%±1.40 | 74.37%±1.33 | 73.22%±1.05 | 89.21%±0.23 | 60.28%±0.26 |
| 100% | 83.60%±0.92 | 73.20%±0.57 | 73.85%±0.94 | 89.67%±0.23 | 60.36%±0.11 | 82.78%±1.91 | 74.83%±1.84 | 73.66%±0.62 | 89.58%±0.52 | 60.39%±0.60 |

Table 7: Accuracies of randomly selecting more data to train the target model. Compared with our selection accuracies, Random needs much more data. 100% YELP and 90% QQP data is necessary for our 20% selection with DistilBERT; 100% YELP and SST2, 80% QQP and AGNEWS are necessary to match our 20% budget performance of BERT.

