# OpenReview forum: "SELECTFORMER: PRIVATE AND PRACTICAL DATA SELECTION FOR TRANSFORMERS"
_ICLR.cc/2024/Conference — Submitted to ICLR 2024_

### Official Review · Reviewer_8Z9X · 2023-10-23

**Soundness:** 3 good
**Presentation:** 4 excellent
**Contribution:** 3 good
**Rating:** 8
**Confidence:** 2

**Summary:**

In the free data market, model owner would like to trade data from data owner to be able to maximize the model accuracy, which requires to select and appraise portions of data points. However, the data selection should be private to both model and data owner in order to keep model parameters and data points private. A technique named MPC can be utilized to jointly evaluate the model privately with forward passes while privacy of both parties could be preserved.  Existing MPC approaches could approximate expensive nonlinearity with cheaper operations for transformer inference, which transformer models are applied in the most deep learning tasks. Meanwhile, they suffers from considerable runtime overhead and poor selection utility. In this work, authors propose an approach to accelerate MPC-based private data selection while preserve the utility of the selection.

**Strengths:**

1. The assumption for the unlabeled and probably imbalanced dataset D from data owner is a practical consideration, which makes the work more convincing and challenging than prior related works.
2. Multiphase selection is a smart strategy. Instead of selecting appropriate data at once. Filtering out most irrelevant data points in the early stage is a way more efficient approach with smaller models and larger models in the later phases provide accurate selection based on filtered data.
3. Approximation of nonlinear operations with multiple shallow MLPs can simulate the utility of nonlinearity well and reduce high dimensions into much lower dimensions to provide huge efficiency improvement.

**Weaknesses:**

1. While a semi-honest setting is convenient and efficient for the research purpose, a setting against malicious adversary is more practical. The assumption based on faithful protocol execution by both parties is not quite strong in the real-world application.
2. The reveal of target model architecture could be dangerous. For example, an adversary can potentially retrieve parameters of certain layer. Instead, I think the target model could be trained with MPC protocols.

**Questions:**

1. Since the setting of model and data owner is based on two parties. Instead of MPC, have you considered to solve this problem in the zero-knowledge proof setting? ZK approach allows model owner to select and appraise the data from data owner by privately committing parameters and data, and providing verifiable results without data leakage. It may be more efficient than MPC (not a conclusion and probably interesting for the future investigation).
2. In the threat model of Section 2.1, can you explain more about what can be revealed (like the purchase budget B mentioned in the Section 4.1) and what should be private (like model parameters and data points)? Or also merge the privacy guarantee in the Section 4.1 into the threat model.
3. What is the threshold of phase i? How should we determine when the early or later phases are?

---

> ### Author Response · Authors · 2023-11-22
> **Response to Reviewer 8Z9X**
>
> We would like to express our sincere gratitude for your review, especially for the positive feedback on the practicality and presentation! We hope that our response has adequately addressed your concerns. If you have any additional feedback concerning our current draft, we would be delighted to hear it and address your points.
>
> We summarize the concerns and answer them individually below:
>
> - Semi-honest setting is not strong enough in real world.
>
> Response: We agree with the limitation of a semi-honest setting. Yet, it is still unsolved and sees active research. We hope the results from semi-honest settings could pave the way to more challenging settings with malicious parties.
> References:
> [1] P. Mohassel and Y. Zhang. SecureML: A system for scalable privacy-preserving machine learning. In IEEE Symposium on Security and Privacy, 2017.
> [2] S. Tan, B. Knott, Y. Tian, and D. J. Wu. CryptGPU: Fast privacy-preserving machine learning on the GPU. In arXiv 2104.10949, 2021.
> [3] P. Mohassel and P. Rindal. ABY3: A mixed protocol framework for machine learning. In ACM Conference on Computer and Communications Security (CCS), 2018.
>
> - The reveal of target model architecture could be dangerous. Instead, train target model with MPC protocols.
> Response: To clarify, revealing model architectures (e.g. layer types, number of layers) does not reveal model parameters, which are considered more sensitive. Training models over MPC would be an interesting approach, which however is challenged by the prohibitive MPC cost.
>
> We answer other specific questions from the reviewer.
>
> **Q1**: Instead of MPC, consider solving this problem in the zero-knowledge proof setting for the future investigation).
>
> **A1**: That’s an interesting direction that we are looking forward to. Besides, we will cite and discuss ZKP in the Related Work section.
>
> **Q2**: Explain more about what can be revealed and what should be private. Or also merge the “Privacy guarantee” into the threat model.
>
> **A2**: Thanks for the suggestion! We have added a new Figure 1 and a “Workflow” paragraph in Section 1 to provide more encryption/reveal information. We intentionally keep “Privacy guarantee” in Section 4.1 to better introduce the multipass selection workflow. Therefore readers will not be confused by encountering these details too early.
>
> **Q3**: What is the threshold of phase i? How should we determine when the early or later phases are?
>
> **A3**: We interpret “threshold” and “early or later phases” as how to decide the number of phases and the dimension of MLPs.
> The number of phases depends on the trade-off between selection performance and computing resources. More phases may have better selection performance but run longer.
> The MLP dimensions in each phase are determined by the schedule via offline grid search(Section 5.4 MLP hidden dimensions, Table 5, and a complete Table 6 in the Appendix).
> For complex datasets, adding one larger dimension phase may increase the selection accuracy (Table 6 in the Appendix).
>
> We hope these additional explanations can address your questions above. We have updated the manuscript for better clarity (new Figure 1, Section 1 Workflow, and our first contribution).

---

### Official Review · Reviewer_due4 · 2023-10-29

**Soundness:** 2 fair
**Presentation:** 2 fair
**Contribution:** 2 fair
**Rating:** 5
**Confidence:** 3

**Summary:**

1. The paper presents a technique called SELECTFORMER for private and practical data selection for Transformers.
2. The contributions of the paper are threefold: (1) a new pipeline for private data selection over MPC, (2) the use of low-dimensional MLPs to emulate high-dimensional nonlinear operators, and (3) a parallel, multiphase scheduling approach for MPC.
2. The goal is to enable the model owner to select and appraise training data from the data owner privately before committing to a transaction.
3. The technique utilizes Multi-Party Computation (MPC) to evaluate the target model over the selected data.

**Strengths:**

1. The research problem is unique and import for practical use.

2. The technique is evaluated on various Transformer models and NLP/CV benchmarks, showing a significant reduction in delay from thousands of hours to tens of hours, with only a minimal accuracy degradation of around 0.20% compared to training with the selected data.

3. The selection process is designed as a multipass sieve, where earlier phases use smaller selector models for quick filtering of redundancy, and later phases use larger selector models for more precise selection.

**Weaknesses:**

1. The first contribution-emulates high-dimensional nonlinearity with low-dimensional MLPs. This is a not novel technique. Besides, training MLPs to approximate every non-linear operator is very cost. Does author evaluate the time cost here?
2. The MPC protocol hide the computation behind the communication data exchange is not novel. Any MPC protocol would try to minimize the total latency like this way. It is not unique here. Thus, I think this contribution is a weak statement.


Reference:

[1] Lu, Lu, et al. "Learning nonlinear operators via DeepONet based on the universal approximation theorem of operators." Nature machine intelligence 3.3 (2021): 218-229.

**Questions:**

See Weakness part.

---

> ### Author Response · Authors · 2023-11-22
> **Response to Reviewer due4 (1/2)**
>
> We would like to express our sincere gratitude for your review! We hope that our response has adequately addressed your concerns. If you have any additional feedback concerning our current draft, we would be delighted to hear it and address your points.
>
> We summarize the questions and answer them individually below:
>
> **Q1**: MLP approximations is not novel.
>
> **A1**: Thank you for pointing out the paper, but this is not applied to our problem because of the goal and the cost. (1) DeepONet wants to solve a different problem from us. It wants to solve the differential equations which will find solutions in input X’s domain. But we want to find an approximation of a function that gives us an output in a different range. (2) Our models have different inputs. Because DeepONet wants to solve differential equations, their inputs consist of both sides of the equation. They require both the left and right-hand side: u(x) and y. However, the only input of our MLPs is just data value x without the right-hand side y. (3) They need more than one big NN which is expensive on MPC. The Trunk net and Branch net have width==40 and depth==3 and 2 respectively. By contrast, our MLP approximation has as low as width==2 and depth==2. Therefore, DeepONet is much more expensive than ours. (Figure 2 has the MPC cost breakdown of each layer). The cost will be even more prohibitive on MPC with the Stacked DeepONet, which has much more Branch net inside.
>
> There is another paper [The-X, ACL2022] mentioned in Section 1 we got inspiration from. We agree that approximation is a well-known trick but the particular design is important, we have revised our draft to clarify (introduction and contribution1).
>
> We will highlight our key discovery: MLP approximation, while cannot be used for generic transformer inference (because of low accuracy), is uniquely suitable for data selection. Such a novel use of MLP approximation is discovered for the first time.
>
> To show that our discovery is non-trivial, we will add the following supporting experiments, which show that MLP approximation, when used in generic inference, results in poor accuracy. Following MPCFormer, we replace three nonlinear modules in a BERT-base model with our approximations: Attention Softmax, Attention LayerNorm, and Feedforward Network Layernorm. There are 3*12 = 36 MLPs in the BERT model. In the experiments, two LayerNorm approximations are always data-aware, while there are both data-aware and fixed approximations of Softmax because of varying attention masks. The BERT with MLP approximation will be fine-tuned on the
> benchmark before inference. Three versions of attention softmax approximations achieve no better than random guess performances:
> (1) Using fixed MLP for attention softmax. Our BERT with approximation achieves only 52.92% accuracy.
> (2) Using data-aware attention softmax approximation, which zeros out masked values on MLP outputs, has 49.08% accuracy. If we normalize the remaining values to [0, 1], the accuracy remains 49.08%.
> To better understand the influence of each nonlinear approximation, we did an ablation study that keeps just one kind of MLP (12 of them) in BERT. However, none of them achieve better than 50.92% accuracy. We further notice that adding just one attention softmax MLP to the model has no impact on the inference accuracy. However, adding one LayerNorm MLP will degrade the accuracy by 0.85% on average. These results show that having just one MLP will hurt the inference performance a lot; having 12 or even 36 MLPs will degrade model performance drastically.
> (3) Removing the attention mask to make MLP data-driven. It always has 50.92% accuracy, no matter with all three approximations or just one approximation.
> The poor inference performance and good data selection performance show that the MLP approximation is specifically suitable for data selection while impractical for model inference directly.
>
> **Q2**: Besides, training MLPs to approximate every non-linear operator is very cost. Does author evaluate the time cost here?
>
> **A2**: MLP training is very cheap actually. They are simple models and we generate data in large batch size==128. The total training time for softmax approximation takes 20.88 minutes for 20K epochs, 5.25 minutes for LayerNorm approximation, and 0.68 minutes for the fuse approximation with the same number of epochs on a Nvidia RTX 2080Ti GPU. It takes only 26.81 minutes of MLP training time in total for one phase, which is cheap compared with tens of hours of selection time.
> Additionally, MLP training should be done before private data selection. Therefore the training cost is even less because of training just once.

---

> ### Author Response · Authors · 2023-11-22
> **Response to Reviewer due4 (2/2)**
>
> **Q3**: The MPC protocol hide the computation behind the communication data exchange is not novel. Any MPC protocol would try to minimize the total latency like this way.
>
> **A3**: To clarify, MPC protocols do NOT specify the order of computation and communication (unless they are dependent); it is the role of MPC frameworks.  To our knowledge, common MPC frameworks such as Crypten do NOT hide computation behind communication automatically.
>
> We will highlight that: our scheduling does not just simply hide computation delays. It has to create such hiding opportunities (by co-executing tasks across the batch boundaries, see Sec 4.4)
>
> We hope these additional explanations can address your questions above. We have updated the manuscript for better clarity (new Figure 1, Section 1 Workflow, and our first contribution).

---

### Official Review · Reviewer_zaeT · 2023-11-01

**Soundness:** 2 fair
**Presentation:** 2 fair
**Contribution:** 2 fair
**Rating:** 5
**Confidence:** 4

**Summary:**

This work proposed an MPC-based private data selection framework for large Transformer models. The main technical contribution of MPC is replacing high-dimensional nonlinearity with low-dimensional MLPs. Besides, a batch evaluation is used in MPC.

**Strengths:**

An MPC-based private data selection framework for large Transformer models.
+ Nonlinearity evaluation with low-dimensional MLPs.
+ Multi-phase selection
+ Parallel MPC executions

**Weaknesses:**

- This work seems to simply combine the techniques of data selection and secure inference on LLMs.
- Replacing high-dimensional nonlinearity with low-dimensional MLPs seems less general.
- The batch evaluation is a widely used method in PPML and lacks novelty.

**Questions:**

1. Is this non-linear evaluation method only applicable in the proposed data selection setting? Can it be extended to general MPC-based LLM?
2. What are the differences between MPC-based data selection on LLMs and secure inference on LLMs?
3. Why do the authors focus on LLMs? Can this method be applied to CNNs?
4. Does this work use a third party for generating correlated randomness for MPC similar to MPCFormer?

---

> ### Author Response · Authors · 2023-11-22
> **Response to Reviewer zaeT**
>
> We would like to express our sincere gratitude for your review! We hope that our response has adequately addressed your concerns. If you have any additional feedback concerning our current draft, we would be delighted to hear it and address your points.
>
> We summarize the concerns and answer them individually below:
>
> - Simply combine the techniques of data selection and secure inference on LLMs.
>
> Response: We need to clarify that we don’t just "simply combine" some techniques. the approximation and proxy model design part is non-trivial. They are efficient in our data selection case, while just borrowing current methods will bring prohibitive private selection costs as mentioned in Section 1.
>
> - Replacing high-dimensional nonlinearity with low-dimensional MLPs seems less general.
>
> Response: There are prior works of approximating nonlinearity with MLP. As mentioned in Section 1, we are inspired by THE-X[ACL2022]. They use MLP approximation in homomorphic encryption. But we improve its efficiency by lowering the dimension, approximating the nonlinear operation as a whole, and data-driven parameters training.
>
> As mentioned by other reviewers, Lu, Lu, et al. "Learning nonlinear operators via DeepONet based on the universal approximation theorem of operators." Nature machine intelligence 3.3 (2021): 218-229. proposes using fully connected network to precisely approximate nonlinear operations for identifying differential equations.
>
> We will highlight our key discovery: MLP approximation, while cannot be used for generic transformer inference (because of low accuracy), is uniquely suitable for data selection. Such a novel use of MLP approximation is discovered for the first time.
>
> We acknowledge that low-dimensional MLPs are NOT for generic transformer inference (New experiment results are in the Appendix). However, We customize it and make it work for an important problem (secure data selection). In other words, we contribute a specialty technique for an important, pervasive problem.
>
> - The batch evaluation is a widely used method in PPML and lacks novelty.
>
> Response: We will highlight that: our system goes beyond standard batch evaluation in PPML, which will result in IO starvation (Sec 4.4). Our system co-executes tasks across batch boundaries in order to create parallelism. At a high level, it brings the job-stealing principle from parallel computing into the context of PPML.
>
> We answer other specific questions from the reviewer.
>
> **Q1**: Is this method only applicable in data selection? Can it be extended?
>
> **A1**: We interpret “LLM” in the question as Transformer models. However, we believe our method can also used on LLMs since they are also transformer-based.
> Yes, it is for selection, not for inference.
>
> **Q2**: What are the differences between MPC-based data selection on LLMs and secure inference on LLMs?
>
> **A2**: Selection has a lower requirement of the proxy model capacity. It only requires the output entropy’s rank to be consistent with that of a full model, but inference needs the proxy model to have the same output logits as those of the full model.
>
> **Q3**: Why do the authors focus on LLMs? Can this method be applied to CNNs?
>
> **A3**: We focus on Transformer models because they have many softmax and LayerNorm modules in each layer, which are very costly for MPC. We didn’t try CNN because they have just a few softmax and LayerNorm modules.
> We believe our method can be extended to CNN models, because our MLP approximation doesn’t rely on model architectures.
>
> **Q4**: Does this work use a third party for generating correlated randomness for MPC similar to MPCFormer?
>
> **A4**: In principle, our design needs no third-party assistance. In particular, the Beavor Triples needed by us can be generated by oblivious transfers offline, without any third party.
>
> We hope these additional explanations can address your questions above. We have updated the manuscript for better clarity (new Figure 1, Section 1 Workflow, and our first contribution).

---

### Official Review · Reviewer_ER9t · 2023-11-04

**Soundness:** 3 good
**Presentation:** 2 fair
**Contribution:** 3 good
**Rating:** 6
**Confidence:** 3

**Summary:**

In this paper, the authors proposed a method to approximate transformer models for training using MLP instead of non-linear functions such as Softmax and layernorm.

**Strengths:**

It seems that the performance of the Softmax function is important in transformers when MPC or FHE is considered. The precision of Softmax approximation has a very large impact on the overall inference performance. In this paper, they suggested to use MLP instead of them.

**Weaknesses:**

The core ideas proposed in the paper are described on pages 4 and 5, but the description in this part is somewhat unclear. In particular, it is unclear whether data transfer between proxy models occurs using MPC in the multi-phase selection section, or not. This part needs to be restated more clearly.

**Questions:**

In the multipass selection phase, it said, "The forward pass computes the prediction entropy values, which are encrypted." The question is, what kind of encryption scheme is used? Is it a homomorphic encryption? How can you generate secret shares from the encrypted entropy values for the input of the following proxy model?

**Details Of Ethics Concerns:**

I have no concerns

---

> ### Author Response · Authors · 2023-11-22
> **Response to Reviewer ER9t**
>
> We would like to express our sincere gratitude for your positive review! We hope that our response has adequately addressed your concerns. If you have any additional feedback concerning our current draft, we would be delighted to hear it and address your points.
>
> We summarize the concerns and answer them individually below:
>
> - Whether data transfer between proxy models occurs using MPC in the multi-phase selection section.
>
> Response: We add Figure 1 to give a better workflow overview. We will clarify that: data is never moved during selection over MPC. Data indices will be transferred. Bootstrap data and selected data will be sent out in the clear before and after selection.
>
> We answer other specific questions from the reviewer.
>
> **Q1**: What kind of encryption scheme of encrypted entropy is used? Is it a homomorphic encryption?
>
> **A1**: It’s not homomorphic encryption. In fact, the entropy, just as other intermediate states, is encrypted as arithmetic and binary secret shares, which are the MPC’s design choices [2, 3]
> [2] I. Damgård, V. Pastro, N. Smart, and S. Zakarias. Multiparty computation from somewhat homomorphic encryption. Cryptology ePrint Archive, Report 2011/535, 2011. https:// eprint.iacr.org/2011/535.
> [3] O. Goldreich, S. Micali, and A. Wigderson. How to play any mental game or a completeness theorem for protocols with honest majority. In STOC, pages 218–229, 1987.
>
> **Q2**: How can you generate secret shares from the encrypted entropy values for the input of the following proxy model?
>
> **A2**: At the very beginning of the selection process, the secret shares of the input data are created; subsequently, all the secret shares (including entropy) are derived through MPC computations. Then next phase will know data indices, not entropy value. Thanks to this suggestion, we have added a new Figure 1 for clarification.
>
> We hope these additional explanations can address your questions above. We have updated the manuscript for better clarity (new Figure 1, Section 1 Workflow, and our first contribution).

---

### Official Review · Reviewer_Xsq1 · 2023-11-07

**Soundness:** 2 fair
**Presentation:** 1 poor
**Contribution:** 2 fair
**Rating:** 3
**Confidence:** 3

**Summary:**

This paper explores the efficient acquisition of data necessary for training artificial intelligence models using multi-party computation techniques. To effectively train models at a fixed cost when acquiring data from data owners, it is necessary to assess the quality of the data. This paper presents a method for data appraisal, allowing model owners to select data that is advantageous to them without accessing the data themselves.

**Strengths:**

The approach of efficiently purchasing data for training artificial intelligence systems within a fixed budget is a novel research direction. This seems to be a necessary research topic not only for AI security but also for various AI training scenarios. The paper successfully persuades the need for the research direction and research topic.

**Weaknesses:**

It seems to lack a clear and precise explanation of the technical aspects of the paper. All the technical details regarding the research method are quite ambiguous, making it difficult to understand the core ideas of the paper. While the paper mentions proposing data selection and appraisal methods when training artificial intelligence models using MPC, it does not precisely explain how these techniques are related to MPC protocols and security. Additionally, it doesn't provide a clear explanation of why each specific technique is necessary and what problems they aim to solve, making it hard to grasp the reasons behind the use of these techniques.

Furthermore, the paper does not offer a clear protocol or source code to understand how each operation is exactly performed. The technical aspects of the paper remain ambiguous, as there is no clear protocol or algorithmic explanation. (I believe that if the protocols and algorithms are given clearly, code submission itself can be considered optional.) The paper lacks explanations for the figures, making it challenging to understand their meanings.

Therefore, I think this paper is not yet ready for presentation at a conference, and a complete rewrite is necessary to clarify each algorithm and protocol and make it more reader-friendly.

**Questions:**

Some specific areas of ambiguity include:

1. How each operation is computed using MPC.
2. How entropy is calculated if MPC is used.
3. How the index with the highest entropy is jointly found.
4. The definition and details of the QuickSelect algorithm.
5. The interpretation of Figures 1 and 2.
6. The role and precise definition of the proxy model.
7. The dimensions and hyperparameters used for the MLP that replace the nonlinear function.
8. The data used for proxy model training.
...

There are many other unclear aspects that need to be clarified to give the paper more value.

---

> ### Author Response · Authors · 2023-11-22
> **Response to Reviewer Xsq1 (1/2)**
>
> We would like to express our sincere gratitude for your review! We hope that our response has adequately addressed your concerns. If you have any additional feedback concerning our current draft, we would be delighted to hear it and address your points.
>
> We summarize the concerns and answer them individually below:
>
> - Lack a clear and precise explanation of the technical aspects.
>
> Response: Thank you for your feedback. Yet, we appreciate it if your comments can be more specific.
>
> - Technical details regarding the research method are ambiguous.
>
> Response: In Section1 Goal & techniques, we explain the design goals, the techniques, and the benefits that our methods can bring. In Section 1 Workflow, we further explain how our methods work and how each technique executes from a high-level perspective.
>
> - Not precisely explain how these techniques are related to MPC protocols and security.
>
> Response: In Sections 2 & 4, we give assumptions and explanations of MPC details. Note that we do not deviate from protocols implemented by well-known frameworks such as Crypten [Crypten, NeurIPS2021].
>
> - Lack the explanation of necessities and targeting problems of techniques.
>
> Response: In Section 1, we talk about the challenges of private selection over MPC and explain the motivation of each contribution. In our Goal & techniques on page 2, we explicitly give the problems we want to tackle for each technique and the benefit it can bring.
>
> - No clear protocol and source code.
>
> Response: We interpret “protocol” in the question as a generic description, not specific to the MPC protocol. Section 1 provides a clear overview of our system operation. (i.e. “protocol”). We interpret “source code” in the question as pseudo-code. Our innovation is a new proxy model architecture and selection workflow. Hence. figures are more suitable in this case.
>
> - No clear protocol or algorithmic explanation.
>
> Response: In Section 1, we provide a clear overview of our system operations. In terms of MPC protocol, we mentioned it in Section 2.1. In Section 2.2 and Section 4, we explain our machine learning algorithm in detail.
>
> - Lack explanations for the figures.
>
> Response: We explicitly provide the message we want to convey in each figure’s title.
> In Section 1 Workflow, we have explanations for Figure 1. In Section 2.3 Major overheads, we have explanations for Figure 2. In Section 4.2 and 4.3, we have explanations for Figure 3 and Figure 4. In Section 5.2, we have explanations for Figure 5 and 6. In Section 5.4, we have explanations for Figure 7.

---

> ### Author Response · Authors · 2023-11-22
> **Response to Reviewer Xsq1 (2/2)**
>
> **This Response (2/2) is identical to the deleted Response (2/2). We deleted that one since we want to directly respond to Reviewer Xsq1, not to our Response (1/2). Sorry for the confusion.**
>
> We answer other specific questions from the reviewer.
>
> **Q1**: How each operation is computed using MPC.
>
> **A1**: In Section 2.1 Threat model, we mention that all operations are supported by Crypten [Crypten, NeurIPS2021]. Before MPC operations, all data must be first encrypted. This includes both the model parameters as well as the inference data. On a high level, they can be seen as matrices. After encryption, the encrypted matrices still have the same shape as before the encryption, but all the entries now contain randomized values. Furthermore, during the MPC operations (addition and multiplication), the data entries remain randomized, but their shapes are still publicly revealed. It’s until both parties decide to share their respective data share to each other and add them up, that the data is decrypted.
>
> **Q2**: How entropy is calculated if MPC is used.
>
> **A2**: In Section 4.3, we mention that entropy is predicted by an MLP. Besides, entropy can also be calculated by the formula, they are logarithms and multiplications supported by Crypten.
>
> **Q3**: How the index with the highest entropy is jointly found.
>
> **A3**: We use QuickSort (Section 4.1 Multipass selection) to find the index with the highest entropy. But since MPC doesn’t support control flow, we reveal the comparison results between two data entropy. Since our goal is to find the index with the highest entropy, this revealing doesn’t hurt the privacy assumption.
>
> **Q4**: The definition and details of the QuickSelect algorithm.
>
> **A4**: It is a textbook algorithm, c.f. Hoare, C. A. R. (1961). "Algorithm 65: Find". Comm. ACM. 4 (7): 321–322. doi:10.1145/366622.366647.
>
> **Q5**: The interpretation of Figures 1 and 2.
>
> **A5**: In Section 4.2, we have a detailed explanation of our algorithm and these two figures (They are Figures 3 and 4 now).
>
> **Q6**: The role and precise definition of the proxy model.
>
> **A6**: In Section 2.2(2) Proxy models, we cite prior works of creating lightweight proxy models for data selection and mention we tailor new proxy model architectures for our goal. In Section 4.2, we explain how to generate proxy models in detail, including their architectures and training process.
>
> **Q7**: The dimensions and hyperparameters used for the MLP that replace the nonlinear function.
>
> **A7**: In Figure 4 and Section 5.4 MLP hidden dimensions, we mention that dimensions are 2/8/16 for a three-phase selection.
>
> **Q8**: The data used for proxy model training…
>
> **A8**: In Section 4.1 Pre-selection bootstrap, we explain that the model owner will use the bootstrap data S_boot to generate proxy models $M^^_{1..N}$.
>
> We hope these additional explanations can address your questions above. We have updated the manuscript for better clarity (new Figure 1, Section 1 Workflow, and our first contribution).

---

### Author Response · Authors · 2023-11-22
**General Response**

We thank the reviewers for their valuable feedback. We are encouraged that they appreciate our paper for the following reasons:
- The problem of doing private data selection over MPC for Transformer models is important and realistic.
- The approach is well-motivated.
- The framework is accurate and efficient.
- Evaluations are thorough and show strong effectiveness.

We summarize some of the main changes to the paper below.

- A new Figure 1 illustrating the three stages in our data selection workflow.
- In Appendix 7.1, Experiment results and explanations of using our MLP approximation in Transformer model inference.
- Our contribution of proposing an MLP approximation technique that is uniquely tailored for data selection.
- Wording and formatting such as extra spaces.

We have responded to each reviewer individually.

---

### Meta-Review · Area_Chair_zx34 · 2023-12-06

**Metareview:**

This paper studies private data selection in transformer model. There are two parties, the model holder, who has a transformer model, and the data holder, who has an unlablled dataset. The model holder wants to pay and buy a subset of the data holder's dataset. However, the model wants to make sure that this subset is as useful as possible, i.e. they want to pick a subset of predefined size such that these can help improve their training accuracy as much as possible. We want to do this selection in a private manner in the two-party semi-honest setting. A standard technique is to build a small proxy model and pick only the points with high entropy in the prediction (indicating less certainty) [Coleman et al., 2019]. Doing this in MPC is expensive because even small transformer model has many non-linear operations. The main contributions in this paper is to propose two improvements: (i) use a multiphase selection where we use small proxy model to weed out most of the datapoints first before moving on to use a large proxy models to get the final selections, and (ii) replace non-linear layers with trained multi-layer perceptrons (MLPs). The latter makes the operations linear which can be implemented much more efficiently in MPC.

## Strengths

- Private data selection seems like a relevant problem and this paper provides significant practical improvements.

- Multiphase filtering is a nice idea that seems to be novel in the private data selection context.

- The fact that MLP approximation works for data selection is quite surprising given that it doesn't work that well for inference.

## Weaknesses

- Lack of clarity in writing: The models and protocol descriptions are written in a very dense manner (and in words, without appropriate notations) and it is hard to follow for any non-expert.
  - As an example of the above, it is not even clear whether, in the multiphase training, the intermediate sets / rankings are leaked.

- Lack of discussion regarding privacy concern given that that ranking is leaked. (Not just the final selected set.)

- Lack of discussion between the usages of MLP in previous work and this work: Previous work uses MLP for *inference* and fail but somehow this work uses it for *filtering* and succeeds; it is unclear why this is the case. (Originally this was barely discussion at all. Now, Appendix 7.1 is added to briefly discuss this but this question remains.)

**Justification For Why Not Higher Score:**

As stated above, I think the paper is poorly written without appropriate formalism on the model and the privacy leakage, so it is not up to par with the standard of conferences like ICLR.

**Justification For Why Not Lower Score:**

N/A

---

### Decision · Program_Chairs · 2024-01-16

Reject